



# Bolchem: an On-Line Coupled Mesoscale Chemistry Model

Rita Cesari[1], Alberto Maurizi[2,3], Tony Christian Landi[2], Massimo D'Isidoro[4], Mihaela Mircea[4], Felcita Russo[4], Piero Malguzzi[2], and Francesco Tampieri[2]

[1]CNR ISAC, Institute of Atmospheric Sciences and Climate, Strada Prov.le Lecce-Monteroni Km 1,200, 73100 Lecce
[2]CNR ISAC, Institute of Atmospheric Sciences and Climate, Via Gobetti 101, 40129 Bologna
[3]CNR IMM, Institute for Microelectronics and Microsystems, Via Gobetti 101, 40129 Bologna
[4]ENEA, Air Pollution Laboratory, Via Martiri di Monte Sole 4, 40129 Bologna

**Correspondence:** Rita Cesari (r.cesari@isac.cnr.it)

**Abstract.** This work presents the on-line coupled meteorology-chemistry transport model BOLCHEM based on the hydrostatic meteorological BOLAM model, the gas chemistry module SAPRC90, and the aerosol dynamic module AERO3. It includes parameterizations to describe natural source emissions, dry and wet removal processes, as well as the transport and dispersion of air pollutants. The equations for different processes are solved on the same grid during the same integration step, by means of an operator splitting method. This paper describes the model and its performance on the European domain for one year run (December 2009-November 2010). The results show that BOLCHEM reproduces both gaseous concentration ($O_3$ and $NO_2$) and particulate matter concentration ($PM_{2.5}$ and $PM_{10}$) at surface. For $O_3$, we found the best agreement in summer, with a correlation coefficient R of 0.72 and mean bias of 4.78. On the contrary, $PM_{10}$ and $PM_{2.5}$ are better reproduced in the winter, the latter with a correlation coefficient R of 0.66 and the mean bias MB of 0.50.

## 1 Introduction

Traditionally, Meteorology and Air Quality (AQ) developed as separate disciplines. From the modelling point of view, this led to development of separate meteorological and Chemical Transport Models (CTM), the latter using as input data the output of the former.

However, over the last decades, it was recognized that the interaction among processes in the atmosphere could not be confined within those two domains. Actually, some of the missing coupling in the offline representation of the atmosphere proved to be fairly important, not only for climate modelling where the impact of greenhouse gases and aerosol on radiation and clouds is of paramount importance in the determination of the global (and local) budgets, but also for short time forecasts of both meteorological and composition quantities, in which aerosols play a major role (see Baklanov et al. (2014) for a review).

Despite this, the development of integrated models of atmospheric dynamics and composition is relatively recent: for instance, the first stable version of WRF-Chem, one of the most widespread models, has been released only in recent years (Grell et al., 2005) and involves a strong coupling of the two "realms" with raising scientific and computational complexities. For these reasons, CTMs, i.e. offline models that typically use the output of a meteorological model to compute transport, diffu-





sion, and transformation of pollutants, are still widely used (see, e.g.,Copernicus Atmospheric Monitoring Service-Europe ( https://atmosphere.copernicus.eu)).

In recent years, large efforts have been undertaken to develop not only integrated models, but also strategies and frameworks for online integrated modelling (see, e.g., Baklanov et al. (2016)) and their evaluation, as, for instance, the joint European –

North American Air Quality Model Evaluation International Initiative (Im et al., 2015a, b).

BOLCHEM is an online coupled mesoscale meteorology-chemistry model where, as defined in Baklanov et al. (2014), equations for different atmospheric processes are solved "simultaneously" (using an operator-splitting technique) over the same grid using one main time step for integration.

The processes that constitute the meteorological component (flow dynamics, thermodynamics, radiation, surface and other

physical processes) are based on the BOLAM model, which is one of the few meteorological models developed in Italy. BOLCHEM development started on 2002, making it one of the first projects for the development of online-coupled models in Europe. The project documentation is available on-line (http://bolchem.isac.cnr.it/data/docs/butenschoen-bolchem_manual-2003.pdf). BOLCHEM was included in the NetFAM/COST 728 database among the "few" online models (see Baklanov et al. (2007)). The chemistry and meteorological components are one-way coupled (online-access, according to Baklanov et al.

(2014)). From 2005 to 2009, BOLCHEM participated to the GEMS Project (Hollingsworth et al., 2008), the Regional Air Quality (RAQ) subproject with a version that included gas chemistry and related processes (Huijnen et al., 2010; Zyryanov et al., 2012), and to MACC - Modelling Atmospheric Composition and Climate. A first study with this BOLCHEM version can be found in Mircea et al. (2008). In the CityZen project, BOLCHEM contributed to the activities on regional climate (Colette et al., 2011, 2012; Maurizi et al., 2012), including aerosol and its interaction with radiation. The model, with the inclusion

of both gas and aerosol chemistry, was applied for specific air quality studies, mainly in the frame of INTERREG ERESIA, INTERREG CESAPO, and MED POSEIDON projects (Cesari et al., 2014; Buccolieri et al., 2016; Cesari et al., 2018).

The aim of this paper is to describe the meteorological and chemical modules of the model (section 2) and to evaluate its performances on the European domain (section 3).

## 2   Model description

Every existing integrated meteorological and chemical transport modeling system (Met-Chem Model) is built on top of a pre-existing meteorological model. This fact typically produces a bias in the model description outline. Here we start description keeping in mind the online-integration framework, outlining the model description according to processes rather than to component basis: turbulence (including dispersion of tracers), interface (including dry deposition), radiation (including primary effect), cloud (including wet removal and - not yet implemented - "aerosol secondary effect"), and so on. The present version

of BOLCHEM is based on the hydrostatic meteorological model BOLAM (Buzzi et al., 2003), the gas-phase chemical mechanism SAPRC90 (Carter, 1990) and the aerosol model AERO3 (Binkowski and Shankar, 1995; Binkowski and Roselle, 2003) and includes a number of parameterizations to describe transport and dispersion, natural emission sources, dry and wet removal





processes and their interactions. These different components have been intimately connected by unifying as much as possible the process involved. This section describes in detail the model components.

## 2.1 Boundary layer and vertical diffusion

The surface layer (SL) is modeled according to the classical Monin-Obukhov similarity theory (Monin and Obukhov, 1954).
The Businger (see Fleagle and Businger (1980)) stability functions are used in the unstable SL, while Holtslag (Beljaars and Holtslag, 1991) functions apply to the stable case. The roughness length over land, initially defined depending on vegetation and sub-grid orographic variance, is modified as a function also of snow coverage conditions. Over the sea, a Charnock roughness (Charnock, 1955) representation is introduced for computing momentum fluxes. It takes into account the dependence of wave height on the surface wind speed, while roughness lengths for temperature and humidity in stable and unstable conditions are defined according to Large and Pond (1981). The mixing length (ML) based turbulence closure, widely used to compute the PBL fluxes for atmospheric modeling (see, for instance, Cuxart et al. (2006)) is applied to model the turbulent vertical diffusion of momentum, potential temperature and specific humidity in the free atmosphere. The turbulence closure is of order 1.5, in which the turbulent kinetic energy (TKE) equation is integrated in time (Zampieri et al., 2005). To take into account buoyancy effects in case of saturated atmosphere, the ML depends on the Richardson number based on equivalent potential temperature. In the unstable case, a modified version of the Bougeault and Lacarrere (Bougeault and Lacarrere, 1989) ML is applied while, in the stable case, a modified Blackadar (Blackadar, 1962) formulation is used. Finally, the dissipated TKE is fed back into resolved temperature (frictional heating).

## 2.2 Convection and precipitation

The sub-grid convective precipitation is treated in BOLCHEM following the Kain-Fritsch (KF) parameterization scheme (Kain and Fritsch, 1993; Kain, 2004) that has shown considerable success in simulating the development and evolution of convection under a variety of atmospheric environments (Wang and Seaman, 1997; Ferretti et al., 2000). The parameterization has been completely re-coded and some modification has been introduced in the original code. Liquid water static energy (instead of the Bolton approximation of equivalent potential temperature) is used as thermodynamic conserved quantity. Moreover, additional modifications have been introduced with respect to the Kain (Kain, 2004) version, regarding the dependency of downdraft on ambient relative humidity, and the precipitation rate. The cloud depth threshold establishing the onset of shallow convection has also been increased. The above changes tend to slightly reduce, on average, the temperature at low tropospheric levels around and below cloud base, hence stabilizing a little more efficiently the lower troposphere. This has also the effect of reducing to some extent the intensity of small-scale cyclogenesis in the presence of convection.

The scheme describing the physical processes of stratiform precipitation is based on a single-moment, bulk microphysics scheme with two ice precipitation categories (snow and graupel). The spectral properties of clouds and hydrometeors are described assuming generalized gamma function distributions. The main processes described by the microphysical scheme are: nucleation of cloud water (cw) and of cloud ice (ci), condensation and evaporation of cw, freezing of cw, sublimation and melting of ci, auto-conversion of cw and of ci into rain and precipitating ice categories, sublimation of snow and graupel,





evaporation of rain, melting/freezing of hydrometeors, hydrometeor and cloud interactions (collection/accretion/riming), computation of terminal fall speeds and fall process, using a conservative backward-upstream integration scheme, thermodynamic feedback based on enthalpy conservation.

Aerosols and gases removal by clouds and precipitation is treated as in the EMEP/MSC-W model (Simpson et al., 2012).
The online availability of the meteorological fields allows a better computation of the effects of spatial and temporal variability of clouds and precipitation on pollutant removal, since the evolution of single cell storms have typical lifespan less than one hour that cannot be properly captured by offline models.

## 2.3  Gas chemistry

The atmospheric gas-phase chemistry is simulated with the SAPRC90 photochemical mechanism, as described in Carter
(1990). This mechanism considers 35 chemical species that undergo 131 reactions. It is designed for an efficient representation of complex ambient mixtures with the purpose of assessing differences in atmospheric impacts of individual VOCs. The role of VOCs in ozone and secondary organic aerosol formation and evolution has still to be clarified. SAPRC90 mechanism includes reactions for a large number of VOCs that have been evaluated against environmental chamber data. The SAPRC90 photolysis rates used in photolytic reactions are calculated by the model as a function of the actual radiation (see section 2.5).
The mechanism includes the gas-phase reactions of $SO_2$, but not heterogeneous reactions. SAPRC90 was extended to describe the formation of condensable organic products and the oxidation of $SO_2$ in aqueous, following the scheme reported in Silibello et al. (2008).

## 2.4  Aerosol dynamics

Chemical composition of aerosol is simulated with the aerosol dynamic model AERO3 (Binkowski and Shankar, 1995;
Binkowski and Roselle, 2003) coupled with the inorganic, thermodynamic equilibrium model ISORROPIA (Nenes et al., 1998), and with the secondary organic aerosol model SORGAM (Schell et al., 2001). The aerosol dynamic model is coupled with the gas-phase chemical model allowing the use of updated gas-phase precursor concentrations. Aerosol variables and number and mass concentration of aerosol species are transported using same time steps and algorithms used for the gas-phase species. The implemented aerosol dynamic module follows the so-called modal approach proposed by Whitby (1978),
in which the population of particles is described by a superposition of log-normal distributions called modes. The aerosol population is described by three modes (Aitken, accumulation, coarse), the aerosol particles are formed by nucleation and growth by condensation and coagulation, both intra- and inter-modal. Gas/particle mass transfer continuously modifies the chemical composition of aerosol. The considered aerosols are: primary unspeciated anthropogenic aerosol and water, primary organic aerosol, primary elemental carbon, soil and sea salt, secondary biogenic and anthropogenic organic aerosol, ammonium, ni-
trate, and sulfate. The amount of ammonium, nitrate, sulfate and water contained in the aerosol is calculated with ISORROPIA (Nenes et al., 1998) and depends on available concentrations of ammonia, nitric acid, $SO_2$ and on relative humidity and temperature. The amount of secondary organic aerosol (SOA), both biogenic and anthropogenic, is calculated using the absorptive partitioning model of Pankow (1994) , extended by Odum et al. (1996), and the concentrations of reacted ROG's, the total (gas





+ aerosol) concentration of each semi-volatile organic compound and the concentrations of primary and secondary organic aerosols.

## 2.5 Radiation

Radiation fluxes are computed with a combined application of the Geleyn scheme (Ritter and Geleyn, 1992) and the ECMWF
cycle 26 scheme (Morcrette, 1991; Mlawer et al., 1997), with both the Tegen (Tegen et al., 1997) aerosol Aerosol Optical Depth (AOD) climatology or the actual single level Aerosol Optical Thickness (AOT) derived from the model aerosol concentration.

The Geleyn scheme, with the option of maximum cloud coverage, is called approximately every 0.5 hours, and has been modified to take into account explicit cloud concentration. The ECMWF scheme is used to correct the surface and internal radiative fluxes of the Geleyn scheme. It is computed every 1.5 hours at alternate horizontal grid points to spare computational
time. Surface fluxes of visible and infrared radiation are then converted into one-time step increments to obtain a smooth time evolution of surface temperature and turbulent fluxes of heat and moisture. Local cloud fraction is parameterized in terms of explicit cloud water/ice content, corrected with a function of relative humidity to account for subgrid effects.

In the original ECMWF scheme, the climatological AOD is partitioned vertically among the model layers using a parameterization based on the temperature profile, therefore calculating the single cell AOT for the radiation calculation. BOLCHEM can
optionally include the actual (modeled) aerosol content in the computation of AOT for the aerosol components included in the Morcrette (1991) scheme: sulfates, organics, sea salt, black carbon. The calculation uses the modelled aerosol concentrations as input and is based on an approximated calculation for the Mie scattering theory (Van de Hulst, 1981). The AOT calculation includes also the humidification of the aerosol by calculating a wet diameter and a wet refractive index following the Hanel formula (Tombette et al., 2008). The AOT module can be adapted to different aerosol schemes since the AOT is calculated for
the single species. The aerosol species for which the AOT calculation has been implemented here are those included in the AERO3 module, grouped in the following classes: sulfates, organics, sea salt, black carbon, ammonium, nitrate. In case aerosol components are simulated only in the lower troposphere, the climatological values of AOT are used above. A preliminary test has been performed to verify the consistency of the AOD variations with the effect on the radiation budget and the impact on the temperature distribution. Results show that the effect on surface temperature is not directly correlated to the surface radiation
flux, but depends also on the vertical distribution of the aerosols. Preliminary comparisons with AERONET measurements give very encouraging results (Russo et al. (2010), https://meetingorganizer.copernicus.org/EGU2010/EGU2010-8561.pdf). However, the impact of aerosol on radiation is not included in the stable version of the model, whose performance is presented in this paper.

Concerning photolytic reactions, the knowledge of photolysis rates requires a rather expensive computation from actinic flux
and molecular properties that cannot be performed run-time. Typical simplified implementations (see, e.g. Kumar et al. (1995)) rely on an offline computed lookup table that reports photolysis rates computed under clear sky conditions as a function of the solar zenith angle. Then, the clear sky values are corrected from optical properties. In contrast to parameterizations developed for offline coupled models that need to diagnose indirect parameters such as the optical thickness of clouds (see, e.g., Chang et al., 1987), the availability of modelled radiation allows for its direct use. In BOLCHEM the photolysis rates are modified





locally (in all grid cell) according to the ratio between the actual radiation and the clear sky radiation, both available from the Morcrette (1991) model. This allows for a 3D spatial distribution of photolysis rates that accounts for variations of the actual local radiation fluxes relative to their clear-sky values. The ratio can exceed 1.0 above clouds and is reduced within and below clouds. Although the correction is applied without reference to specific spectral properties of different molecules, it can

roughly capture the overall modification of the photolysis activity without adding heavy computation. Considering the present overall errors in atmospheric integrated models, the use of sophisticated radiation schemes for photolysis computation seems not fully justified in practical applications.

## 2.6 Surface fluxes

Surface fluxes (sources and sinks) are the most critical input for a Met-Chem. Moisture and heat influences atmospheric dy-

namics while anthropogenic and natural emissions are the source of gas and particulate pollutants. On the other hand, removal by dry deposition can be an important sink for them. In the following, submodels of surface fluxes for the different components are described. While anthropogenic contribution can be only included from available databases, natural components sources (heat, moisture, gases and aerosols) couples different components such as radiation, turbulence, soil type and moisture content. Different parameterisations are used for different processes. According to the online approach, all of the modules used for the

calculation of surface fluxes of the various components share all the available information. Mineral dust is not considered here because is part of a subproject of BOLCHEM, DREAMABOL (Binietoglou et al. (2015); Terradellas et al. (2018), Northern Africa-Middle East-Europe (NA-ME-E) Regional Center, https://sds-was.aemet.es/), that will be treated separately.

### 2.6.1 Anthropogenic emissions

The anthropogenic emission data are based on European Database or National Database. In the present work, we use the

TNO-MACC-III emission database (available for 2000–2011), that is an update from the TNO-MACC-II dataset (Kuenen et al., 2014). It provides gridded emissions on the European domain, at 0.125° x 0.0625° longitude-latitude resolution, for each year, country, sector and source type. Eight key families of pollutant are considered: methane ($CH_4$), carbon monoxide (CO), ammonia ($NH_3$), non-methane volatile compounds (NMVOC), nitrogen oxides ($NO_x$), coarse particulate matter ($PM_{10}$), fine particulate matter ($PM_{2.5}$), sulfur dioxide ($SO_2$). The annual emissions are then split in time to hourly resolution. NMVOC

and PM are speciated on the basis of emission sectors and seasonal activity (Monforti and A., 2005), according to the chemical module SAPRC90 and the aerosol module AERO3. Vertically, the emissions are allocated on three levels (0m, 50m, 150m). For the point source emissions, it was used a plume rise scheme (Hanna and Briggs, 1982).

### 2.6.2 Soil and vegetation model: moisture and heat

The meteorological component BOLAM includes an original soil model with fixed lower boundary conditions, where heat

and water vertical transfer are computed at the interfaces of several soil layers with depths ranging typically from a few cm to more than 1 m, increasing downward. Vegetation effects at the surface (transpiration and interception of precipitation)





and in the soil (extraction of water by roots depending on wilting conditions), taking into account different soil types and physical parameters, are considered. The soil model also includes a treatment of freezing and melting processes of the water content. At the surface, the evolution of the snow cover is computed considering snow accumulation and melting, with a single layer snow mantle model. A surface skin temperature is defined by imposing zero-net flux divergence of heat at the

soil-atmosphere interface. Albedo and emissivity variations are also computed as a function of the uppermost soil water content. The sea surface temperature is predicted using a slab ocean model, where latent and sensible heat fluxes, and radiation contributions, are taken into account. The soil and vegetation dataset is obtained from the Global Land Cover Facility of the University of Maryland at 1/120 degree resolution (Hansen et al., 2000), a classification from AVHRR satellite and free available (ftp://ftp.glcf.umiacs.umd.edu/glcf/Global Land Cover/Global).

### 10  2.6.3   Biogenic gas emissions

Surface flux of visible radiation and skin temperature, calculated run time by the BOLCHEM model at every time step, are used to estimate the biogenic gas emission fluxes. Biogenic emissions are grouped into three categories: isoprene ($C_5H_8$), monoterpenes ($C_{10}H_{16}$) and OVOCs ($C_xH_yO_z$). Following Guenther et al. (1995); Simpson et al. (1999), the biogenic emission flux is given by the product of the emission potential, the foliar biomass density and an environmental correction factor representing

the effects of temperature, and in some case solar radiation, on emissions. The emission potential and the foliar biomass density are ecosystem dependent. BOLCHEM uses an independent data base having 1 km spatial resolution (Symeonidis et al., 2008): a non-canopy approach is adopted and the emission potential is calculated at branch-levels. For monoterpene, two separated emission potentials are provided. Depending on the plant specie, the monoterpene emission flux is driven by temperature only, or by light and temperature. The correction factor is calculated following Guenther et al. (1993). For isoprene, as for monoter-

pene light and temperature depending, the factor is the product of two coefficient which account respectively for the effect of leaf temperature and Photosynthetically Active Radiation (PAR). PAR is calculated in BOLCHEM as a function of the surface flux of visible radiation. For monoterpene and OVOCs emission from most plant, the correction factor is a factor depending on temperature only (Guenther et al., 1993, 1995). BOLCHEM uses the skin temperature to calculate the correction factor.

### 2.6.4   Sea salt

Sea salt emission flux is calculated following Zhang et al. (2005). In this approach, the solute weight fraction of seawater, composed in the model by $Cl^-$, $Na^+$ and $SO_4^{2-}$, is represented as a function of RH (Tang et al., 1997) over the 0.45-0.99 RH range. Then, the density of a sea-salt particle is expressed as a function of the solute weight fraction of sea water. The surface fluxes of sea-salt are first calculated at the reference relative humidity of $80\%$, using an open-ocean source functions that empirically relate the particle number size distribution and the wind speed at 10m a.s.l. ($U_{10}$) (Monahan et al., 1986; Gong,

2003). This function is applicable for $U_{10} < 20ms^{-1}$ and $0.8m < r_{80} < 10m$, being $r_{80}$ the wet particle radius at $RH = 80\%$. The size distribution of sea salt particle flux is then adjusted according to the local relative humidity, using a correction factor, expressed as a function of ambient relative humidity RH. The wind speed $U_{10}$ and the relative humidity RH are calculated by BOLCHEM at every time-step.





### 2.6.5 Forest fire emissions

Forest fire emissions can be estimated for particular case studies and are included in the model (Cesari et al., 2014). The pre-processor *prebolchem_fire* is described in Pizzigalli (2012). Wildfire emission fluxes are estimated starting from latitude and longitude of area burned obtained from satellite products, as MODIS burned area product, following the methodology proposed
by Seiler and Crutzen (1980). The emission E(x) of specie x is given by the product of the burnt area (A), the fuel load (B), the burning efficiency CE and the emission factor e(x) of species x. B and e(x) are function of the land cover classification and CE is a function of the tree cover. For each class of vegetation, values of fuel load and burning efficiency are estimated for each class of vegetation, referring to the UMD Global Land Cover Classification (GLCC) (Hansen et al., 2000). The WRAP approach (WRAP, 2005) is used to modulate the emission fluxes and to estimate the fire emission height. Basically, the fires are
classified into size classes based on virtual acreage, then top and bottom emission height ($Htop_{max}$ and $Hbot_{max}$) and buoyant efficiency $BE_{size}$ are assigned to every plume class. The total fire emissions are split into the smoldering and flaming part. The smoldering fraction is calculated as a function of the BE size and it is daily modulated as a function of the BE size and buoyant efficiency. The remaining part is distributed between the bottom and the top of the plume following a diurnal profile, with peak emissions during early afternoon and low emissions during the night.

### 2.6.6 Dry deposition

Parameters from the BOLAM surface layer, such as roughness length and stability functions (see section 2.1) temperature, pressure, soil and vegetation types are directly provided at every time step to the gas dry deposition module. The scheme is based on a resistance analogy approach (Wesely, 1989) in which the dry deposition velocities can be written as the inverse of the sum of three resistances, $Vd = (R_a + R_b + R_c)^{-1}$, where the three quantities $R_a$, $R_b$ and $R_c$ are the aerodynamic,
quasi-laminar and canopy resistances, respectively. The formulation of $R_c$ follows the methodology implemented in Anav et al. (2012). It uses the parameterization of Emberson et al. (2000) to account for stomatal conductance, that is calculated for different land cover types as a function of the vegetation type, phenology, and surface parameters like temperature and humidity. The algorithm is adapted to the BOLCHEM soil and vegetation classes, using 16 vegetation and 9 soil types.

### 2.7 Numerical core

The model prognostic variables are distributed in the vertical on a non-regular Lorenz grid (Lorenz, 1960), with higher resolution in the atmospheric boundary layer near the lower surface. The vertical discretization is based on a hybrid vertical coordinate system, in which the terrain-following coordinate gradually tends to a pure pressure coordinate with increasing height above the ground. The horizontal discretization is based on a staggered Arakawa-C grid (Arakawa and Lamb, 1977), in geographical coordinates (latitude-longitude). The equator can be taken to be any great circle on the Earth in order to minimize
grid anisotropy over the limited area of the simulation. The time scheme of BOLAM is split-explicit, forward-backward for the gravity modes. The lateral boundary conditions are applied on a number of grid point frames, using a relaxation scheme





(Leheman, 1993) that efficiently absorbs wave energy, helping in reducing spurious reflection by the lateral boundaries. The SAPRC90 equations are solved by the `lsode` solver with adaptive time-step.

The advection scheme presently implemented is the Weighted Average Flux scheme (WAF) (Billet and Toro, 1997), which is a second order implementation of the Godunov (1959) method particularly suited to integrate in time the conservation

of a scalar quantity. This scheme is a 'total variation diminishing' one, and therefore prevents the occurrence of spurious oscillations (see also Hubbard and Nikiforakis (2003)). The horizontal velocity components are interpolated over the T-points of the Arakawa C grid, advected as any other variable defined on those points, and interpolated back to velocity points (to avoid smoothing, one-dimensional fourth order interpolation is used).

Conservation of mass of advected and diffused species is easily implemented in an online model. In order to conserve mass

exactly in the transport of the scalar parameters involved in the chemical core, a flux-form of the BOLAM advection scheme has been used in which the advected quantities are expressed as volumetric concentrations (where the volume is defined with the terrain-following coordinate).

Initial and boundary conditions for meteorology can be taken either from the European Centre for Medium range Weather Forecasts (ECMWF) Integrated Forecasting System (IFS) or from the National Centers for Environmental Prediction (NCEP)

Global Forecasting System (GFS). Lateral boundary conditions are imposed using a relaxation scheme (Davies, 1976; Leheman, 1993), while a sponge layer is adopted at the top of the atmosphere, which is fixed to be at $\sigma = 0.01$, corresponding to about 10hPa (middle stratosphere). Initial and boundary conditions for chemical species can be taken from different sources depending on the availability, including climatological fields. Boundary concentrations are injected into the model by means of an inflow/outflow condition applied to the external grid frame. In addition, to save CPU time, the top level for chemistry can

be optionally chosen to be lower than that for meteorology.

## 2.8   Model configuration

BOLCHEM has been used in different applications, from experimental forecast over Europe (GEMS) and Italy (MACC) to long term (climatological) studies (CityZen). In these applications, the horizontal resolution ranges from 50 to 10 km with a number of vertical levels ranging from 30 to 60. Time discretization depends on horizontal resolution and is defined using

the rule-of-thumb: $dt \sim dlon * 1000$, with dlon expressed in degrees. Apart from specific studies where the full troposphere is simulated (Aidaoui et al., 2015), the chemistry/aerosol components are typically simulated only from ground up to $0.5\,\mathrm{sigma}$ that roughly corresponds to 5500m a.s.l. The meteorological component can use boundary conditions (BC) from ECMWF (IFS) and NCEP (GFS), the latter being used typically in forecast mode for real time applications. As far as the atmospheric composition is concerned, boundary conditions can vary depending on the specific application. In the GEMS and MACC

projects, output from global and regional forecasting systems, respectively, were used. However, the number of species and vertical levels provided by such "services" are very limited. Experimental forecast was set up at ISAC where the boundary conditions for a simulation at 10 km resolution over Italy were provided by a nesting of BOLCHEM into itself, with the parent simulation covering the whole Europe with climatological BC. Results of the model nested configuration are shown in Cesari et al. (2019).



The meteorological component of BOLCHEM uses the hydrostatic approximation and thus the horizontal resolution is limited to about 5-6 km. This represents a limitation to reproduce pollutants dispersion in some context where local circulation and emission sources are highly variable over shorter scales.

Anthropogenic emissions must be provided by the user for the specific case considered. A major limitation in the actual im-
plementation is that the biogenic emissions are potential available only over Europe (Simpson et al., 2012). However, following the same methodology, it is possible to build the information for other areas of interest. In addition, in order to better integrate the different components (meteorology, chemistry, vegetation) in a fully coupled on-line model, it is preferable to connect the biogenic emissions to the vegetation database already used by the meteorological component.

At present, no meteorological feedback is routinely included in calculations. Feedback on precipitation requires a more
sophisticated scheme of hydrometeors. Feedback on radiation has been implemented only in an experimental form.

As BOLCHEM was built merging different codes with different settings policies, and because running the whole model requires the control of a large number of (heterogeneous) parameters that sometimes appear with different names in different parts of the ensemble of setting, the model was equipped with a system that unifies the run setting and allows to manage the whole run, from the definition of the domain (space-time) and resolution to the download of meteo ic/bc, preparation of
emissions, and the run itself. Every parameter is accessible to the user in a unique $run.ini$ with extensive description that provides a sort user-guide. The BOLCHEM-run system is actually a collection of tools managed by a python script and relies on tt cfget, a free software specifically developed for the purpose (https://tracker.debian.org/pkg/cfget).

## 3   Model evaluation

### 3.1   Model setup

The assessment of model performance was done by using a one-year run over Europe (December 2009 - November 2010), with a horizontal resolution of about 40 km. A spin-up period of 30 days (November 2009) was used and not considered for model verification. Boundary conditions for the meteorological processor were taken from ECMWF analysis every six hours. Meteorological runs were initialized every day at 00 UTC, while atmospheric composition was carried through the whole simulation, i.e, the concentration after one-day simulation was used as initial condition for the next day run. The chemical
module (both aerosol and gas phase) are driven by a climatology provided by INERIS, in the frame of the CityZen project (Colette et al., 2011). Anthropogenic emissions are based on the TNO-MAC-III emission database for the year 2010. The model setup follows the standard configurations in terms of PBL parameterization, microphysics, as reported in sections 2.1 and 2.2. Ground measurements were obtained through the public database of the European Environmental Agency AIRBASE (https://www.eea.europa.eu/data-and-maps/data/airbase-the-european-air-quality-database-2), which contains air quality data
delivered annually under 97/101/EC Council Decision establishing a reciprocal exchange of information and data from networks and individual stations measuring ambient air pollution within the Member States.

This experiment aims to verify the model skill in capturing the main features of the major air quality indicators at continental and seasonal scales. Seasons are defined as follows: winter (DJF) includes December (2009), January and February; spring





(MAM) includes March, April and May; summer (JJA) includes June, July and August; autumn (SON) includes September, October and November. The model evaluation is focused on the $NO_2$, $O_3$, $PM_{2.5}$ and $PM_{10}$ concentrations at surface level for background stations. Model surface concentration is interpolated bi-linearly at the AIRBASE locations of the selected stations. Hourly concentrations of $O_3$ and $NO_2$ and daily concentrations for particulate matter $PM_{2.5}$ and $PM_{10}$ were considered over

the European region (-15E to 30E; 30N to 60N).

In order to give a general view of the model results, with the purpose to point out the model capability in reproducing the spatial variability over the European domain, seasonal average of observed values superimposed to modeled ones over the domain of interest are presented in Figure 1. Seasonal average of concentration for particulate matter $PM_{2.5}$ and $PM_{10}$ and $NO_2$ is considered during the cold season (DJF), while seasonal average of $O_3$ concentration is considered during the warm

season (JJA). According to measure availability, daily values for $PM_{2.5}$ and $PM_{10}$ and hourly values for $NO_2$ and $O_3$ are used. In Fig. 1, the position of used Airbase stations is represented by circle marks. At the same locations, filled circle represent the measured concentrations.

## 3.2   Model results

Figure 1 depicts a fairly good overall agreement, with well captured spatial structure. In particular for $O_3$, higher concentration

values are found in the South Regions of the domain and over the Mediterranean Sea. Values over Spain and the French Mediterranean coastline are well reproduced, with a slight overestimation over Northern Italy (Po Valley). For $NO_2$, high value over Po Valley are well reproduced, concentration values over North Europe and Great Britain regions are overestimated. For $PM_{2.5}$ and $PM_{10}$ we found a reasonably good overall agreement. In particular, in the most critical areas over the European domain, such as Po Valley and Benelux, the model captures the observed, strong spatial gradient. The high concentration values

observed over Poland are also well reproduced.

The model also captures the monthly variability, as shown in Fig.2. Monthly mean value is considered for all the pollutants, with the exception of $O_3$, for which the monthly mean of maximum daily concentration is considered. The larger model underestimation is observed for $O_3$ during the cold season, while we found a good agreement with observations for summer, when the $O_3$ levels are higher. For $NO_2$ the larger overestimation is observed in winter period. For $PM_{2.5}$ the model well

reproduces the monthly variability throughout the year with a weak overestimation during spring.

More information on model capability to describe $O_3$ behavior, in particular the daily variability, can be obtained by the diurnal cycle, which is mostly driven by the solar radiation. $O_3$ in summer presents a typical diurnal cycle with maximum located in the early afternoon, which is the main feature affecting air quality. Figure 3 shows, for the summer season, the diurnal cycle of the $O_3$ measured concentration vs that calculated by the model. The model well reproduces the diurnal cycle

of ozone concentration.

Model performance in reproducing the observed concentrations, without any averaging operation (temporal or spatial), is highlighted by the scatter plot diagram. Fig. 4 display concentration data as hexagonal points, each having as x-coordinate the measured pollutant concentration [$\mu g/m^{-3}$], and as y-coordinate the simulated pollutant concentration [$\mu g/m^{-3}$] (at the same time and the same Airbase station). In the color bar are reported the number of samples falling in each hexagon. Hourly values



**Figure 1.** Seasonal average of model (maps) and measurements (filled circles) of $O_3$ concentration $[\mu g/m^{-3}]$ for JJA season (a), $NO_2$ (b), $PM_{2.5}$ (c) and $PM_{10}$ (d) concentration for DJF season.

have been used for $NO_2$ and $O_3$ concentrations, and daily value for $PM_{2.5}$ and $PM_{10}$. Results are shown on seasonal basis. For $O_3$, the best agreement is found in the summer. For the spring season, also affected by a moderately high $O_3$ concentration, the model performances are satisfactory, although a slight model underestimation is evident. More persistent underestimation is present for the cold seasons, especially in winter. This result doesn't represent a serious limit to model application, being $O_3$ not a typical winter pollutant. On the contrary, winter is often affected by high concentration of $PM_{10}$, that are well reproduced by the model. The model underestimates high concentration values and overestimates the lower. Encouraging results were achieved for the remaining seasons, as pointed out by the basic statistics reported in Tab. 1. $PM_{2.5}$ shows results similar to those of $PM_{10}$. $NO_2$ shows similar results for all seasons, with a slight model overestimation of low concentration values and underestimation of high concentration values. It is evident the good model performance achieved in simulating the $O_3$ concentration during the warm season and the particulate matter ($PM_{2.5}$ and $PM_{10}$) during the cold season. For these pollutants, the best agreement was then found in the seasons where the two pollutants are most abundant, respectively. It is





**Figure 2.** Average values calculated overall background stations for whole period of interest, from December 2009 to November 2010, for O$_3$ (a), NO$_2$ (b), PM$_{2.5}$ (c) and $PM_{10}$ (d). Daily mean values are reported through blue filled circles (observations) and red empty circles (model). For O$_3$, the mean of maximum daily concentration is considered. In addition the monthly mean with solid line is represented (red line represents simulated value, while blue line represents measured values).

worth notice that the basic statistics, $R = 0.72$ for O$_3$ as well as $R = 0.66$ for PM$_{2.5}$ underpin the good model performance in terms of temporal correlation.

For providing additional information about the model skill on seasonal basis, Taylor diagrams are shown in Fig. 5. The diagram provides a way of showing how the correlation coefficient R, the standard deviation (sigma) and the (centered) root-



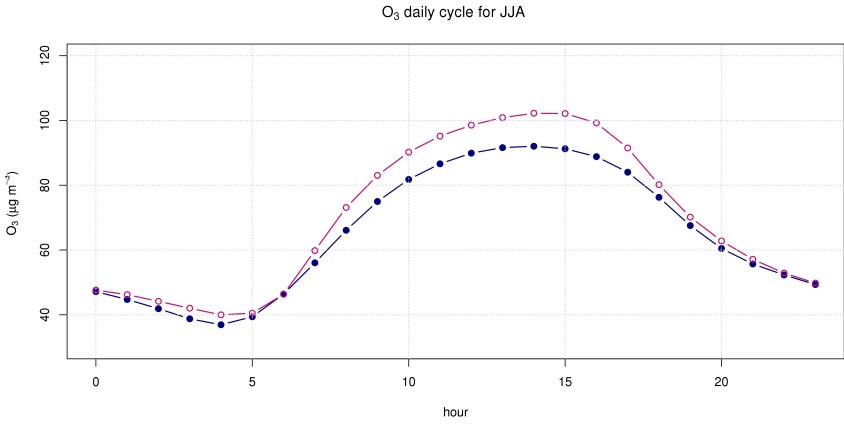

**Figure 3.** Hourly mean ozone concentration averaged overall background station for the summer season. Observations (blue curve) and modeled concentrations (red curve) are reported.

**Table 1.** Basic statistics: correlation coefficient (R), root mean square error (RMSE) and mean bias (MB) calculated for $O_3$, $NO_2$, $PM_2.5$ and $PM_{10}$ on seasonal basis for the background stations.

| | $O_3$ | | | $NO_2$ | | | $PM_{2.5}$ | | | $PM_{10}$ | | |
|---|---|---|---|---|---|---|---|---|---|---|---|---|
| Season | R | RMSE | MB | R | RMSE | MB | R | RMSE | MB | R | RMSE | MB |
| MAM | 0.63 | 25.13 | -4.61 | 0.47 | 20.55 | 2.37 | 0.63 | 9.95 | 3.00 | 0.56 | 13.89 | 0.59 |
| JJA | **0.72** | 26.97 | **4.78** | 0.41 | 18.16 | 1.85 | 0.49 | 7.66 | -0.58 | 0.46 | 12.26 | -4.91 |
| SON | 0.70 | 20.63 | -3.47 | 0.44 | 19.48 | 3.05 | 0.60 | 10.38 | 1.19 | 0.53 | 15.33 | -0.57 |
| DJF | 0.57 | 25.51 | -15.34 | 0.43 | 22.20 | 4.62 | **0.66** | 17.82 | **0.50** | **0.54** | 25.34 | **-0.36** |

mean-square error vary simultaneously. The diagrams point out that for the $O_3$ the best model performances are in summer and autumn, with the higher value for the correlation coefficient R, the lower value for the RMSE and the normalized SD close to unit. Anyway, also for the others seasons, the value of SD indicates that the variability of the model data is similar to the variability of observations. In winter, the model data having less variability than the measurements (SD < 1) and the correlation coefficient has the lower value. For $NO_2$, the model data have more variability than the measurements in all season, except in winter. The correlation coefficient shows similar values in all seasons. Model $PM_{2.5}$ data show more variability in summer season. For $PM_{10}$, Fig. 5 highlight the general underestimation of the model data variability in all the season. The model has best performance in the winter and spring season, and the worst in the summer. It can be observed that for all pollutants the correlation coefficient has typical values found by others CCMM model and that the model performances are close to those achieved by the current state-of-the-art model system dedicated to air quality study, e.g. Im et al. (2015b, a), and Copernicus CAMS products (e.g. Chianese et al. (2018)).





**Figure 4.** Scatter plots for hourly $O_3$ (a) and $NO_2$ (b) concentration and daily $PM_{2.5}$ (c) and $PM_{10}$ (c) concentration. The point refers to the samples having the concentration value $[\mu g/m^{-3}]$ reported on the axis. Such calculations was done on seasonal basis: upper-left spring, upper-right summer, down-left autumn and down-right winter.



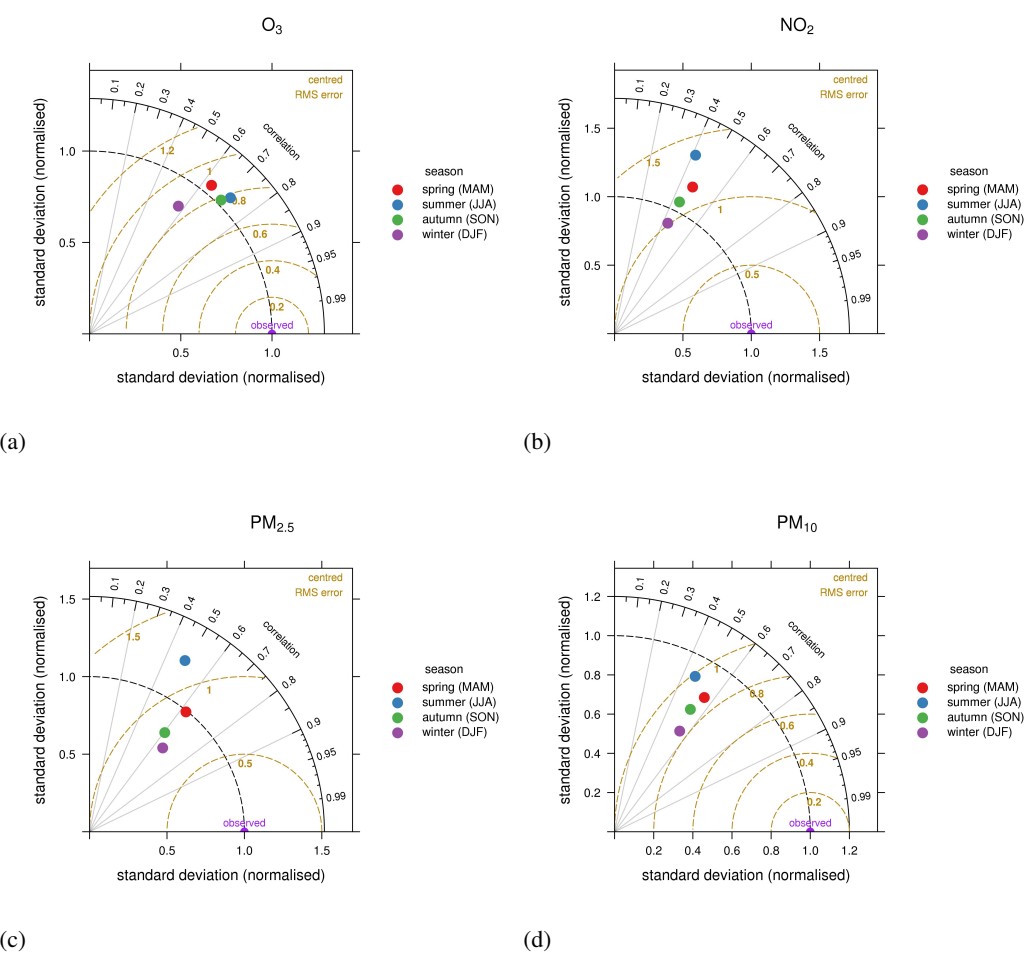

**Figure 5.** Taylor diagram generated for $O_3$ (a), $NO_2$ (b), $PM_{2.5}$ (c) and $PM_{10}$ (d) on seasonal basis as reported in the legend.



## 4 Conclusions

We have presented a description of the on-line air-quality model BOLCHEM that is based on the hydrostatic meteorological BOLAM model, the gas chemistry module SAPRC90 and the aerosol dynamic module AERO3. BOLCHEM has been used for different applications, both forecast and air pollution case studies in the framework of different European projects such as

GEMS and CESAPO. The model was evaluated on seasonal basis (for the year 2010) for $NO_2$, $O_3$, $PM_{2.5}$ and $PM_{10}$, using measured concentrations available from values measured at background AirBase database. $PM_{2.5}$ and $PM_{10}$ are particularly well reproduced by the model in winter time when the $PM_{10}$ concentration usually exceeds the limit values imposed by EU Air Quality Directive on ambient air quality and cleaner air for Europe (AQD, Directive 2008/50/EC). For $PM_{2.5}$ the correlation coefficient ranges between 0.49 (summer period) and 0.66 (winter period). For $PM_{10}$ the correlation coefficient ranges between

0.46 (summer period) and 0.54 (winter period). The model well reproduces the ozone concentration and its daily cycle during summer when it has the highest concentrations and impact on vegetation, in particular on forests. The correlation coefficient of ozone ranges between 0.57 and 0.72, the latter value referred to the summer period. Future developments of BOLCHEM will include:

1. Estimation of emission potential and foliar biomass density referring to the vegetation dataset data set used in BOLCHEM,
that is UMD Global Land Cover Classification (GLCC).

2. Implementation of a pre-processor system to allow the integration of the mineral dust flux simulated by the model DREAMBOL model into BOLCHEM boundary condition.

3. Implementation of the gas-phase chemical mechanism SAPRC-07 (Carter, 2010).

4. A major challenge to complete BOLCHEM aerosol feedback description would be to study the effect of aerosol compo-
nents that are currently not included in the radiation module, such as ammonium and nitrates.

*Author contributions.* Alberto Maurizi and Francesco Tampieri started the BOLCHEM project. Alberto Maurizi coordinated the model development and he is the main developer of the code. Piero Malguzzi is the main developer of the meteorological module BOLAM included in BOLCHEM. Rita Cesari, Alberto Maurizi, Massimo D'Isidoro, Mihaela Mircea, Felicità Russo, and Francesco Tampieri contributed to the developed of the model. Rita Cesari, T. Christian Landi and Alberto Maurizi performed the set up, simulations and analysis for the presented
model evaluation. All the authors discussed the model results. Alberto Maurizi proposed the paper structure. The paper was written by Rita Cesari with contributions by all other authors.

## Code availability

The BOLCHEM model is available upon the signature of an agreement with the CNR-ISAC Dynamic Meteorology Group (contact: r.cesari@isac.cnr.it).





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
