# Peer review of "Bolchem: an On-Line Coupled Mesoscale Chemistry Model"

_Geoscientific Model Development, 2019_

## Short Comment (SC1) · 26 Jul 2019

Dear authors,

in my role as Executive editor of GMD, I would like to bring to your attention our Editorial version 1.2:

https://www.geosci-model-dev.net/12/2215/2019/

This highlights some requirements of papers published in GMD, which is also available on the GMD website in the 'Manuscript Types' section:

http://www.geoscientific-model-development.net/submission/manuscript_types.html

In particular, please note that for your paper, the following requirement has not been

met in the Discussions paper:

- "The main paper must give the model name and version number (or other unique identifier) in the title."

Please add a version number for BOLCHEM in the title upon your revised submission to GMD. Be aware, that exactly the version published and evaluated in this article needs to be permanently archived (and provided to the editor and the reviewers).

Furthermore, provide a reason, why BOLCHEM can not be made publicly available. You are publishing in an open access journal, why not follow open source policy? Finally note, that according to our new Editorial (v1.2) all data and analysis / plotting scripts should be made available.

Yours,

Astrid Kerkweg

---

## Short Comment (SC2) · 6 Aug 2019

Dear Astrid Kerkweg,

the version number of the model will be added; the code, the data and the analysis / plotting scripts will be archived as soon as I arrange with co-authors the suitable archival location.

Best regards, Rita Cesari

---

## Referee Comment (RC1) · Anonymous Referee #1 · 27 Aug 2019

Based on the formulation of the title, it is expected that the paper aims to present the on-line coupled model BOLCHEM.

In the introductory section, the Authors refer to already developed on-line and on-line coupled models with very general references to Baklanov et al. (2014) and Baklanov et al. (2016) without model names and citations. Only the WRF-Chem model is mentioned as "recently released" with the reference from 2005. It should be clear that after 14 years, it is not a "recent" publication or a "recently released" model.

As stated in the Introduction, the model documentation is dated for 2003 (lines: 12-13, page 2). In line 14, it is stated that the model is "one way coupled", which is not consistent with the title of the submitted publication. Further, different publications documenting model applications are cited. It is not clear which version of the model is

presented in the paper.

For the model description section it is suggested to remove the first three sentences (lines: 25-29, page 3) and begin with "The presented version of BOLCHEM . . ." – the information on what is NOT implemented in the presented version is not relevant (line 29, page 3).

Parametrizations of meteorological processes are presented in sections 2.1 and 2.2. However, in the context of this paper, the Authors should include the information on how meteorological processes interact with gaseous and aerosol species. An attempt was made to refer to wet removal in 2.2 section. However, only a reference to the EMEP/MSC-w model was provided. The general statement (lines: 5-7, page 4) can be removed. Description of model dynamics and physics is superficial.

In section 2.4, it was stated, "Aerosol variables and number and mass concentration of aerosol species are transported using scheme time step and algorithms used for the gas-phase species". However, the information on the advection scheme was not provided earlier.

In section 2.5, the Authors try to summarize the radiation process for both the meteorological and chemical components of the model. The presented description is not well organized and does not provide any useful information for the reader. The reference to the ECMWF model cycle 26 (r1 or r3) is from 2003. For a reader not familiar with the history of changes in the ECMWF model, this information is irrelevant. The interactive (coupled) aspect the impact of aerosols on radiation is not included in the "stable version of the model" (line 27, page 5). This statement is inconsistent with the title of the presented publication. The description of a method to estimate photolysis rates is superficial. It is not clear if the method takes into account surface albedo, partial column ozone, or pressure to access the lookup table (or tables). Also, the Authors neglected to specify the spectral regions, absorption coefficients, and quantum yields used for J value calculations. The reference to Kumar et al. (1995) is very dated – over

the last 24 years, there were numerous publications and changes to the spectral data to characterize atmospheric species. Thus, the Authors should provide a reference for the chemical kinetics data used, i.e. JPL Publication No. 15-10, Chemical Kinetics and Photochemical Data for Use in Atmospheric Studies.

It could be suggested to reorder Section 2.6. The "introduction" to this section should be removed (lines: 9-14, page 6). Also, as already indicated, the information on what was not included in the model is not relevant (with two references to the "not included" process).

The emission section should group anthropogenic, biogenic, sea salt and fire emissions, in a separate section.

Vertical allocation of anthropogenic emissions up to 150m is highly questionable. PM10 is not a coarse particulate matter. For biogenic emissions, no details concerning "independent database" were provided (apart from the reference). For sea salt emissions, there is no information on the method used to calculate U10. Description of dry deposition processes is very general.

In Section 2.7, details of horizontal and vertical discretization are mixed with the description of the advection scheme and the treatment of lateral boundaries and chemical solver.

Model configuration section (2.8) contains a list of previous BOLCHEM applications without any references. The grid resolution and number of layers is given as a range (at least a table should be provided). The "rules" to maintain numerical stability related to the Courant number are referenced to as "rule-of-thumb". This statement ("approach") is wrong and not acceptable in a scientific publication. In the description of available chemical boundary condition, the Authors did not include Copernicus products (both in the global and regional scales).

The statement (lines: 4-5, page 10) on unavailability of biogenic emissions outside

[Figure]

Europe is false.

On page 10, lines 9-10 it is stated, "no meteorological feedback is routinely included in calculations". Once again, the statement is contradictory to the title of the submitted paper. Also, it is not clear from the text and presented experiments that the model is run "routinely" in any setup.

Horizontal model resolution of 40km is not sufficient to demonstrate benefits from the on-line approach. Also, it is much coarser than most recently published experiments and analysis. The choice of the year 2010 to present model results for evaluation was not justified.

The analysis is limited to the presentation of qualitative model discrepancies that are described imprecisely – e.g. "a fairly good overall agreement", "a reasonably good overall agreement", "variability of the model data is similar to the variability of observation", "correlation coefficient has typical values found by other CCM models". The presented analysis is limited to the "surface" layer.

Summary:

In the opinion of the reviewer, the presented paper is not suitable for publication in the submitted form.

Specifically:

1. The title is misleading. The title does not reflect model capabilities and the version used for the validation run.

2. The model computer code was not provided to the reviewer.

3. The description of meteorological and atmospheric chemistry sections is not balanced. Description of some of the process is superficial.

4. There is no clear link between atmospheric dynamics and chemistry. The presented model is not coupled (two-way interaction).

5. There are many very general statements on air-quality modelling and on-line models that do not bring any added value to the paper.

6. The description of the presented model suggests that Authors would like to publish the model version that is not "up-to-date" in terms of available emission inventories, boundary conditions, and the state of the science.

7. The model was developed and described in 2003. Model results from different projects were already published most likely with some components of the BOLCHEM that is under the current review. The Authors allude to a coupled version of the model that might exist somewhere.

8. The advantages of the on-line approach were not demonstrated.

9. The evaluation presented in the paper does not demonstrate any usability of the BOLCHEM model for different resolutions, case studies, or regions.

10. The authors presented a one-year simulation for 2010 without any explanation for the selection of the particular simulation period.

11. The presented model evaluation analysis is superficial, and it does not demonstrate any particular advantages for using the BOLCHEM model.

---

## Referee Comment (RC2) · Anonymous Referee #2 · 9 Sep 2019

The article presents the BOLCHEM online coupled model. It is pleasant to read, well structured and understandable. The introduction puts the model and its development back into the global environment of online modeling. BOLCHEM aims to be a state-of-art model, as shown by the part 2, describing in detail the main features of its meteorological and chemical parameterizations. Finally part 3 shows validation simulation performed and comparison against atmospheric concentrations for several pollutants. BOLCHEM model provide good results for both gazeous species and particulate matter. Thus, this paper perfectly fit the scope of GMD journal.

However, more or less serious issues are present in the paper. First of all, the author does not define the version actually analyzed. Nor is it specified whether the model is a global model or a limited area model. In the introduction part, it is said that BOLCHEM

is one of the pioneers of online coupling in Europe without mentioning the others. In general, there are few references to other comparable models.

In the model description, the author says that the development was not done simply by coupling chemistry to a meteorological model, but by taking great care to integrate physical and chemical processes, hence the description of the physical modules that follows. However, it is also indicated that BOLCHEM is based on the BOLAM model, and even (line 14 of the introduction) that it is a "one-way" coupling. This is totally contradictory: a simple quotation from a previous article describing BOLAM is sufficient for meteorological components such as transport or diffusion. Thus, parts 2.1, 2.2 and 2.5 are either to be removed or developed, in particular by indicating the equations of the different parameterizations and how the interactions between chemistry and meteorology are taken into account. As they stand, they are superficial and therefore do not provide any relevant information. Section 2.6 presents how the model manages surface exchanges. Again, the information provided is incomplete. Rather than describing the anthropogenic emissions available in the TNO-MACC-III inventory (one quotation is sufficient), the author should explain how they are distributed over the "3 levels" of the model. Why not include desert dust, which is a major contributor to primary aerosols, in this publication? The reader is often lost between what is really included in the described model and what is not. Part 2.8 (model configuration) is very general and does not provide any relevant information.

Part 3.1 describing the model configuration is incomplete. What is the maximum altitude of the model, the number of levels, how are the conditions adapted to the chemical limits? A table could provide an quick overview to the reader. In addition, the chemical boundary conditions provided by CAMS are, as the author points out, very limited. How are boundary chemical conditions treated for species not provided by the CAMS service? Section 3.2 presents the results of the model. The analysis that is made is too brief. No attempt to analyses the observed biases (underestimation of ozone in winter, overestimation of PM in summer). There is also no analysis of the results of the

upper levels. Finally, it would have been interesting to compare the model results with the CAMS set, which would have been possible by choosing a more recent simulation year.

In conclusion, this article does not currently meet the criteria required for publication